# Internalization of *Salmonella* in Leafy Vegetables during Postharvest Conditions

**DOI:** 10.3390/foods12163106

**Published:** 2023-08-18

**Authors:** Jinnam Kim, Soeun Park, Jiyoung Lee, Seungjun Lee

**Affiliations:** 1Major of Food Science and Nutrition, College of Fisheries Science, Pukyong National University, Busan 48513, Republic of Korea; kjn0032@gmail.com (J.K.); bibi7moon@pukyong.ac.kr (S.P.); 2Department of Food Science & Technology, The Ohio State University, 1841 Neil Avenue, Columbus, OH 43210, USA

**Keywords:** foodborne pathogen, fresh produce, food safety, internalization

## Abstract

The consumption of fresh produce is increasing due to its role in promoting a healthy and balanced diet. However, this trend is accompanied by increased foodborne disease cases associated with pathogens such as *Escherichia*, *Listeria*, and *Salmonella*. Previous studies provided evidence that the internalization of foodborne pathogens in fresh produce may be a potential contamination route and may pose a public health risk. This study investigates the combination effects of storage temperature and humidity on *Salmonella* internalization in six types of leafy greens (iceberg lettuce, romaine lettuce, red lettuce, green onion, spinach, and kale) during the storage stage. The results indicated that temperature plays a critical role in *Salmonella* internalization, with higher concentrations observed in samples stored at 25 °C compared to those stored at 7 °C. The mean concentration of internalized *Salmonella* in the iceberg lettuce sample was the highest and that in the green onion sample was the lowest (iceberg lettuce > red lettuce > romaine lettuce > spinach > kale > green onion). Mist conditions also had an impact on internalization. The group treated with mist showed an increase in *Salmonella* internalization of about 10–30% rather than the group without mist treatment. This research emphasizes the importance of understanding the factors influencing bacterial internalization in fresh produce and highlights the need for proper storage conditions to minimize the risk of contamination and ensure food safety.

## 1. Introduction

The consumption of fresh produce has consistently increased because intaking fresh produce is a key factor of a healthy and balanced diet [1]. The World Health Organization (WHO) recommends about 400 g of fresh produce every day for preventing chronic diseases, including diabetes, cardiovascular disease, and obesity [2]. Concomitantly, accompanying this trend has been an increase in the incidence of foodborne disease cases associated with *Campylobacter, Escherichia coli*, *Listeria*, Norovirus, and *Salmonella* [3]. In particular, *Salmonella* in fresh produce is ranked among the top five foodborne pathogens associated with foods of non-animal origins [4]. *Salmonella enterica* outbreaks associated with fresh produce were reported in the USA and Europe [3,4,5]. These outbreaks accentuate the challenges to the agricultural and food industries as well as to public health.

Previous studies have revealed the ability of foodborne pathogens (e.g., *E. coli*, *Listeria*, *Salmonella*) to interact with the plant, such as attachment, colonization, and internalization by certain plant- or microbial-associated factors [6]. In particular, under laboratory conditions, the internalization of foodborne pathogens in fresh produce may be one of the potential routes for contamination of fresh produce [7,8]. Previous studies demonstrated that foodborne bacteria, including *Salmonella* and *E. coli*, can be internalized in various crops including lettuce, onions, spinach, etc. Two major routes of bacterial internalization into plants are that (1) bacteria can move into plants through physical opening sites in the plant surface (stomata, lenticels, and root) or mechanically damaged areas of the plant surface and (2) bacteria have the ability of motility for penetrating the epidermis of plants [9]. Grivokostopoulos et al. (2022) and Ge et al. (2012) [4,6] found that levels of internalized *Salmonella* in leafy greens were 1.3–4.3 log colony-forming unit (CFU)/g of leafy greens, including arugula, chicory, green onions, lettuce, and spinach. In addition, during the cultivation stage, water stress, such as drought and heavy rain, enhanced *Salmonella* internalization in green onion and lettuce [3]. Therefore, contamination of fresh produce with internalized bacteria shows an increasing risk to public health as produce is mainly consumed. Remarkably, internalized bacteria into the plant are difficult to inactivate effectively by general washing steps, such as tap water or the most common disinfectant (e.g., chlorine and peracetic acid) [10]. Ge et al. (2012) showed no significant reduction in levels of internalized foodborne pathogens in lettuce and green onion treated with chlorine and peracetic acid.

Therefore, to enhance public health, it is necessary to comprehensively understand important factors for the internalization of foodborne pathogens. Previous studies focused on bacterial internalization related to different types of crops, such as lettuce, onion, spinach, etc. Previous studies also demonstrated the internalization of foodborne pathogens in different types of fresh produce during the production process (e.g., seedling and growing). However, it is hard to find the effects of environmental conditions, such as humidity and temperature, on the internalization of foodborne pathogens, such as *Salmonella*. In this study, we focused on *Salmonella* internalization into various crops during the postharvest stage. Practically, most of the fresh produce was kept under various conditions after the harvest stage. For instance, crops were stored at different temperatures with either mist or non-mist, such as the outdoor market and the commercial market. In this study, we investigated the impacts of both temperature (7 °C and 25 °C) and humidity (mist and non-mist conditions) during the storage conditions on *Salmonella* internalization with six different types of leafy greens, including iceberg lettuce, romaine lettuce, red lettuce, green onion, spinach, and kale.

## 2. Materials and Methods

### 2.1. Fresh Produce Samples and Pathogenic Bacteria: Salmonella

Six different types of fresh produce, including iceberg lettuce (*Lactuca sativa* var. *capitata*), romaine lettuce (*Lactuca sativa* var. *longifolia*), red lettuce (*Lactuca sativa* var. *crispa*), green onion (*Allium fistulosum*), spinach (*Spinacia oleracea*), and kale (*Brassica oleracea* var. *sabellica*), were purchased from a commercial market in Columbus, OH, USA.

To distinguish from existing bacterial flora in the samples, green fluorescence protein (GFP)-labeled *Salmonella enterica* serova Typhimurium (ATCC 19585) with an antibiotic-resistant gene (ampicillin) was used in this study [6]. To grow the GFP-labeled *S*. Typhimurium, Luria-Bertani (LB) medium with 100 µg/mL ampicillin (Sigma, St. Louis, MO, USA) was used in a shaking incubator (New Brunswick I2400 Incubator Shaker, Edison, NJ, USA) at 37 °C. The pellet of the GFP-labeled *S*. Typhimurium was collected using centrifugation at 6500× *g* for 10 min and then resuspended using deionized water (Fisher Scientific, WI, USA). The bacterial concentration was determined using the cell density meter (WPA biowave, Biochrom, Cambridge, UK) [6]. A plate count method was also used for bacterial quantification in triplicate [6].

### 2.2. Salmonella Inoculation

GFP-labeled *S*. Typhimurium was cultured in LB broth supplemented with ampicillin (10 µg/mL) for 18 h at 37 °C in a shaking incubator (New Brunswick I2400 Incubator Shaker, Edison, NJ, USA). After making 1 × 10^6^ CFU/mL of bacterial suspension with deionized water, 1 mL of the suspension was contaminated on the surface of the lettuce samples (1 × 10^6^ CFU/leaf). The contaminated samples were stored for 3 days in a sterile container under four different conditions and each group had 10 replicates; (1) at 25 °C with mist, (2) at 25 °C without mist, (3) at 7 °C with mist, and (4) 7 °C without mist (Appendix A). The mist treatment with a sprayer (Stainless Steel Adjustable Nozzle sprayer, model: B07T3J9ZFZ, 3.5 L tank, pressure 2.0 bar, Vevor, UK) was applied for 1 min (0.5 mL per min) at predetermined intervals (every 2 h) and final humidity in the container was approximately 75%. In addition, the lettuce sample without *S*. Typhimurium inoculation was used as a negative control (ddH_2_O) in each group. All experiments were repeated three times.

### 2.3. Removing Bacteria from the Surface of Leaves and Quantification of Internalized Bacteria

To remove the surface bacteria (non-internalized bacteria) in the samples, our previous method was applied [6]. Briefly, the samples were submerged into 80% ethanol (Merck, Darmstadt, Germany) for 10 s and then 1% AgNO_3_ (silver nitrate, Merck, Darmstadt, Germany) for 5 min, and finally rinsed with deionized water (Merck, Darmstadt, Germany) for 10 s. About 5 g of lettuce samples were homogenized with 20 mL of 0.1% peptone water (Thermo Fisher Scientific, Carlsbad, CA, USA) in a sterilized Whirl-Pak bag (Nasco, Fort Atkinson, WI, USA) for 5 min using a stomacher (Stomacher 80, Seward, West Sussex, UK). One hundred microliter of the suspension were spread on LB agar media supplemented with ampicillin (100 µg/mL) in triplicate. The LB agar plates were incubated at 37 °C for 24 h aerobically. Concentrations (colony) of the GFP-labeled *S*. Typhimurium were determined under UV light. The detection limit was 1.0 log CFU/g. In addition, the ampicillin-resistant gene was used to distinguish targeted *S*. Typhimurium from other background bacterial flora. Ten samples for each group were examined, and all experiments were conducted in triplicate.

### 2.4. Imaging Internalized GFP-Labeled Salmonella

To take an image of internalized GFP-labeled *Salmonella* in the sample, the leafy parts of each sample were cut into a piece (area ~1 to 2 cm^2^) of the middle periphery part of each sample using a sterile sampling knife (Fisher Scientific, WI, USA). The samples were laid the piece down on the glass slide with the lower epidermis up for imaging. An upright fluorescence microscope (Axioplan 2, Carl Zeiss, Göttingen, Germany) with a color digital camera (AxioCam MRc Carl Zeiss, Göttingen, Germany) was used to view a flattened area where most stomas could be focused, and then, we took fluorescent images from 6 randomly selected regions for each sample using 20×/0.50 objective lens (Ph2 Plan-Neofluar, Zeiss). We obtained the fluorescent image of GFP-labeled *Salmonella* contaminated leaf using the optical filter sets, especially for the acquisition of FITC fluorescence and operated by Zeiss imaging software (AxioVision, Rel. 4.8.2). The resolution of each image is as high as 1300 × 1030 pixels covering an area about 0.7 × 0.55 mm^2^, i.e., proximally, 1 pixel = 0.53 µm in length or 0.28 µm^2^ in area. All leaf samples were handled at the same time with the same conditions, such as the same illumination in the dark room and same exposure period set by microscope and imaging software (AxioVision, Rel. 4.8.2).

### 2.5. Quantification of Internalized GFP-Labeled Salmonella from the Digital Images

To morphometrically assess the fluorescent area and brightness of GFP-labeled *Salmonella* contaminated in the leaf, we used ImageJ (version 1.52, Fiji, the National Institutes of Health, Bethesda, MD, USA) to quantify the fluorescent area of GFP-labeled *Salmonella* shown in each image by a threshold with a gray level ranging between 190 and 255. We measured and averaged the total pixels of specifically detected GFP-labeled *Salmonella* contaminated area for each digital image, the average the percentage of positive fluorescent area in the total area in each image, and the average gray level of GFP-labeled *Salmonella* contaminated area in each image from 6 randomly selected leaf regions.

### 2.6. Statistical Analysis

Results are expressed as mean ± standard error. The two-way analysis of variance (ANOVA) was applied to determine the differences between different treated groups using SPSS 24.0 software (SPSS Chicago, IL, USA). A *p*-value of <0.05 was considered to be a significant difference with the Tukey HSD post hoc test.

## 3. Results

### 3.1. Effect of Storage Conditions on Salmonella Internalization

To evaluate internalized bacteria only, the surface disinfection steps were conducted and investigated by following our previous study [6]. After the washing steps with ddH_2_O, AgNO_3_, and 80% ethanol solutions, bacteria on the plant surface were not detected (<0 CFU/g of each plant). Concentrations of internalized *S*. Typhimurium in various fresh produce, including iceberg lettuce, romaine lettuce, red lettuce, green onion, kale, and spinach, were determined and presented in Figure 1 and Table 1. Previous studies revealed the internalization of foodborne pathogens (*Campylobacter*, *E. coli*, and *Salmonella*) into different types of fresh produce [6,11,12]. These studies also focused on the internalization of foodborne pathogens during cultivation. However, our study demonstrated a general pattern of *Salmonella* internalization into various types of fresh produce during the storage step. In addition, temperature for storage was a critical factor in *Salmonella* internalization into leaves of each fresh produce. Particularly, under 25 °C with mist condition, the mean concentration of internalized *Salmonella* in the iceberg lettuce sample was the highest and that in the green onion sample was the lowest (iceberg lettuce > red lettuce > romaine lettuce > spinach > kale > green onion). In addition, the common pattern of the concentration of internalized *Salmonella* was similar to the group stored at 25 °C with mist conditions. Furthermore, to investigate *Salmonella* internalization on leaf surface of the fresh produce, a fluorescence microscope was used (Figure 2). From the microscopic images, it is obvious that GFP-labeled *Salmonella* (green) can internalize in fresh produce. The images showed various statuses of GFP-labeled *Salmonella* in the iceberg lettuce, such as underneath the stomata in the substomatal and/or intercellular region, around the rim of the stomata, and in the iceberg lettuce.

#### 3.1.1. Iceberg Lettuce

From day 1 to day 3, mean concentrations (log CFU/g of iceberg lettuce) of internalized *Salmonella* Typhimurium in the iceberg lettuce samples (mean ± SE) were 3.0 ± 0.4 in the 25 °C with mist condition; 3.5 ± 4.5 in the 25 °C without mist condition; 2.2 ± 0.4 in the 7 °C with mist condition; and 1.7 ± 0.3 in the 7 °C without mist condition. Temperature (25 °C and 7 °C) was a significant factor for *Salmonella* internalization (ANOVA, F = 10.4, *p* < 0.05). However, no significant difference of *Salmonella* internalization was found between mist and non-mist groups (ANOVA, F = 0, *p* > 0.05).

#### 3.1.2. Romaine Lettuce

From day 1 to day 3, mean concentrations (log CFU/g of romaine lettuce) of internalized *Salmonella* Typhimurium in the romaine lettuce samples (mean ± SE) were 2.8 ± 0.3 in the 25 °C with mist condition; 2.5 ± 0.3 in the 25 °C without mist condition; 2.0 ± 0.3 in the 7 °C with mist condition; and 1.9 ± 0.3 in the 7 °C without mist condition. Concentrations of internalized *Salmonella* were significantly different between the groups (ANOVA, F = 6.1, *p* < 0.05).

#### 3.1.3. Red Lettuce

From day 1 to day 3, mean concentrations (log CFU/g of red lettuce) of internalized *Salmonella* Typhimurium in the red lettuce samples (mean ± SE) were 2.9 ± 0.4 in the 25 °C with mist condition; 2.5 ± 0.4 in the 25 °C without mist condition; 2.2 ± 0.3 in the 7 °C with mist condition; and 2.0 ± 0.3 in the 7 °C without mist condition. We also found significant differences between the groups (ANOVA, F = 5.0, *p* < 0.05).

#### 3.1.4. Green Onion

From day 1 to day 3, mean concentrations (log CFU/g of green onion) of internalized *Salmonella* Typhimurium in the green onion samples (mean ± SE) were 1.8 ± 0.4 in the 25 °C with mist condition; 1.3 ± 0.4 in the 25 °C without mist condition; 1.3 ± 0.4 in the 7 °C with mist condition; and 1.3 ± 0.4 in the 7 °C without mist condition. Significant differences in internalized *Salmonella* between groups were observed (ANOVA, F = 45.9, *p* < 0.05).

#### 3.1.5. Kale

From day 1 to day 3, mean concentrations (log CFU/g of kale) of internalized *Salmonella* Typhimurium in the kale samples (mean ± SE) were 2.0 ± 0.3 in the 25 °C with mist condition; 1.6 ± 0.2 in the 25 °C without mist condition; 1.5 ± 0.3 in the 7 °C with mist condition; and 1.5 ± 0.3 in the 7 °C without mist condition. The difference between internalized *Salmonella* concentrations was significant between the groups (ANOVA, F = 55.1, *p* < 0.05).

#### 3.1.6. Spinach

From day 1 to day 3, mean concentrations (log CFU/g of spinach) of internalized *Salmonella* Typhimurium in the spinach samples (mean ± SE) were 2.2 ± 0.4 in the 25 °C with mist condition; 2.0 ± 0.3 in the 25 °C without mist condition; 1.7 ± 0.3 in the 7 °C with mist condition; and 1.7 ± 0.3 in the 7 °C without mist condition. There was a significant difference in the *Salmonella* internalization between the groups (ANOVA, F = 8.2, *p* < 0.05).

### 3.2. Effects of Temporal Variation on Salmonella Internalization

To investigate the effects of temporal variation on internalized *Salmonella* after internalization into fresh produce, concentrations of *Salmonella* were measured for 3 days (Figure 1, Table 1). Concentrations of internalized *Salmonella* had decreased over time, except the iceberg lettuce samples (iceberg lettuce: F = 0.9, *p* > 0.05; romaine lettuce: F = 14.2, *p* < 0.05; red lettuce: F = 18.3, *p* < 0.05; green onion, F = 14.4, *p* < 0.05; kale: F = 9.9, *p* < 0.05, spinach: F = 49.9, *p* < 0.05). A previous study demonstrated that concentrations of internalized bacteria gradually increased over time (2~15 min) when fresh produce was immersed in contaminated water [13,14]. However, in this study, fresh produce was contaminated with *Salmonella* at one time without additional contamination, and then trends of internalized *Salmonella* until 3 days were investigated.

### 3.3. Quantification of Internalized GFP-Labeled Salmonella from the Digital Images

Using the fluorescence microscope with ImageJ software (version 1.52), fluorescent areas (%) of GFP-labeled *Salmonella* were measured to estimate levels of internalized *S*. Typhimurium (Figure 2 and Figure 3). The general pattern of fluorescent area (%) was similar to the data generated from the plate counts (CFU), except for the green onion and kale samples. Temperature and mist conditions were important factors in bacterial internalization into fresh produce. The groups stored at 25 °C showed a larger fluorescent area (%) than the groups stored at 7 °C. In addition, mist treatment facilitated bacterial internalization into fresh produce, except for kale and spinach samples.

## 4. Discussion

Contamination of pathogenic bacteria on fresh produce from pre-harvest sources (soil, irrigation water, fertilizers, and wildlife) and post-harvest sources (harvesting process, washing and processing facilities, transportation, and retail and consumer handling) may pose a major public health risk, especially to vulnerable populations, including young children, elderly individuals, pregnant women, and immunocompromised individuals [15]. In particular, the internalization of pathogenic bacteria into fresh produce is a critical issue because internalized bacteria become hard to eliminate through conventional removing methods, such as washing and rinsing [6,16]. Chemical sanitizers also have limited effectiveness in combating these internalized bacteria [6]. With UV irradiation (6000 J/m^2^), internalized levels of *Salmonella* and *E. coli* were reduced by 0.8 to 2.4 log CFU/leaf [9]. Furthermore, internalized bacteria can survive and/or multiply within the plant tissue under favorable conditions, such as nutrients, pH, moisture, and temperatures [17,18]. Previous studies emphasized that understanding bacterial internalization into fresh produce with environmental factors is important for establishing effective prevention and control strategies [19,20].

Many studies have focused on bacterial internalization into fresh produce from germination to cultivation [6,9,21,22]. Ge et al. (2012) and Gu et al. (2013) demonstrated that internalization of *Salmonella* species in lettuce (2–3 log colony forming unit (CFU)/g) and tomato (0–3 log CFU/g)) occurred in seedling and mature plants [6,22]. Park et al. (2013) examined internalization of *E. coli* in spinach during a preharvest period [21]. In addition, they found internalized *E. coli* persisted over 20 days and the level of internalized *E. coli* increased 400 times. Previous studies emphasized various factors that contribute to the internalization of pathogenic bacteria in fresh produce across all stages of production, handling, and storage to mitigate the public health impact of foodborne diseases [6,21,22].

In this study, the effect of storage conditions (temperature and mist condition) on the internalization of pathogenic bacteria (*S*. Typhimurium) was studied in different types of fresh produce (iceberg lettuce, romaine lettuce, red lettuce, green onion, spinach, and kale. The mean concentration of internalized *S*. Typhimurium differed for each type of fresh produce due to the density and size of the opening sites, such as stomata and damaged areas, on the leaves [9,13]. Furthermore, surface morphology, chemical and biological tissue composition, and metabolic activities of each fresh produce may impact bacterial internalization. For example, kale can produce a high amount of glucosinolate, which is related to the enhancement of plant growth and defense mechanisms against plant diseases [9]. The present study reveals that temperature (25 °C vs. 7 °C) plays a crucial role in internalization of *S*. Typhimurium in fresh produce. Temperature differentials (−5.6 °C vs. 5.6 °C) significantly affected bacterial internalization in plants [23]. Temperature may affect the adhesion of bacteria to the surface of fresh produce [24]. Generally, higher temperatures can facilitate the binding of bacteria to the surface of fresh produce, especially when combined with humidity [25]. The previous study found that the initial attachment of bacteria on the surface of fresh produce can lead to the formation of colonization and to the enhancement of internalization [26]. Furthermore, temperature impacts defense mechanisms of fresh produce, either strengthening or weakening the defenses depending on the types of fresh produce and specific bacteria [27]. Temperature also leads to increased plant transpiration, which facilitates the internalization of bacteria into the plant tissue through natural openings, stomata, or damaged areas [28]. Temperature affects growth, survival, motility (ability of bacteria to move), and chemotaxis (ability of bacteria to follow a chemical gradient) of bacteria which can influence the internalization process [26]. For instance, at optimal temperatures (37 °C), *Salmonella* species increase their ability to migrate towards openings on the surface of fresh produce for internalization [1]. Water plays a significant role in the internalization of pathogenic bacteria into fresh produce by providing a means of transport for pathogens to enter the internal tissues of the plant [29]. Commonly, high moisture levels cause stomata and other plant tissue openings to stay open for longer periods [29]. In addition, water contributes to the survival and growth of bacteria on plant surfaces [30]. The length of time spent in storage stages contributes to the growth and survival of internalized bacteria [5,6,7]. Without proper sanitation and temperature controls, extended periods in these stages can result in increased growth of bacteria [5]. In this study, we could not find significant differences from day 1 to day 3 because 25 °C and 7 °C temperatures are unfavorable temperatures for growth of *Salmonella*.

This study investigated the effects of temperature and moisture during storage on the internalization of *Salmonella* in fresh produce. We also focused on only *Salmonella* internalization into various fresh produce, including iceberg lettuce, romaine lettuce, red lettuce, green onion, kale, and spinach. However, previous studies have demonstrated impacts of bacterial internalization on strains of bacteria such as *E. coli* and *Salmonella* because of the nature of each bacterium, indicating a need for further research.

Additionally, analyzing changes in the freshness of fresh produce during a three-day storage period requires further investigation and a study is needed to investigate the impact of *Salmonella* internalization on crop freshness. Finally, this study on bacterial internalization was conducted in laboratory settings, so an epidemiological study is necessary.

## 5. Conclusions

Our present study examines the levels of internalized foodborne bacteria in various fresh produce with different environmental conditions, such as temperature and mist conditions. Understanding the process of bacterial internalization in fresh produce is essential for devising effective strategies to minimize the risk of foodborne illnesses associated with the consumption of fruits and vegetables. Our findings contribute to the knowledge base on this topic and highlight the importance of a comprehensive approach to address the challenges posed by internalized pathogens in fresh produce.

## Figures and Tables

**Figure 1 foods-12-03106-f001:**
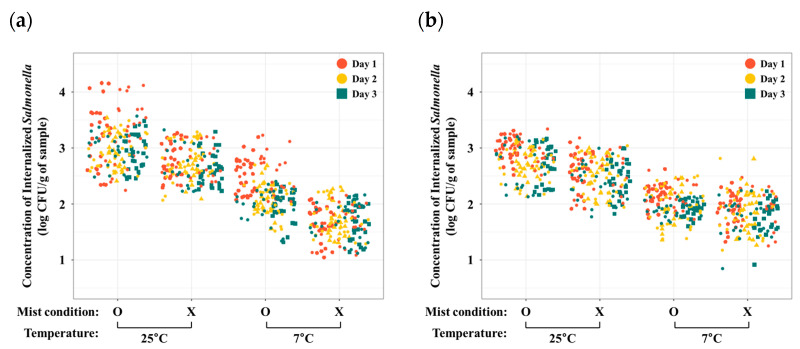
Concentrations of internalized *S.* Typhimurium (log CFU/g of sample) in iceberg lettuce (**a**), romaine lettuce (**b**), red lettuce (**c**), green onion (**d**), kale (**e**), and spinach (**f**) from day 1 (red circle), day 2 (yellow circle), and day 3 (green square).

**Figure 2 foods-12-03106-f002:**
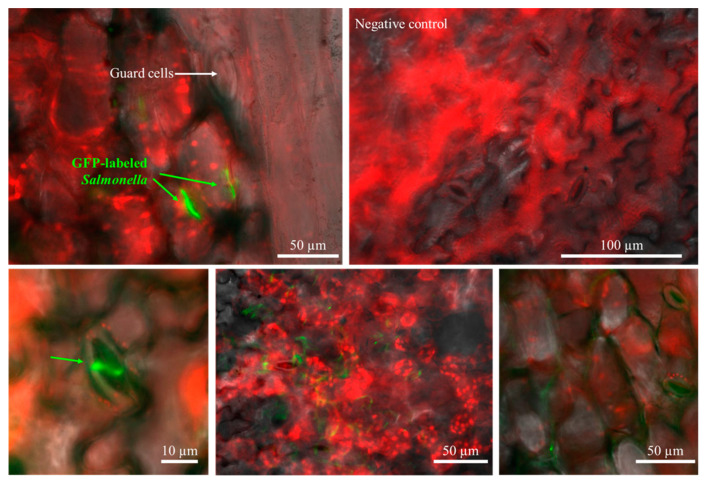
Microscopic images showing the topography of the surface of romaine lettuce and GFP-labeled *S.* Typhimurium.

**Figure 3 foods-12-03106-f003:**
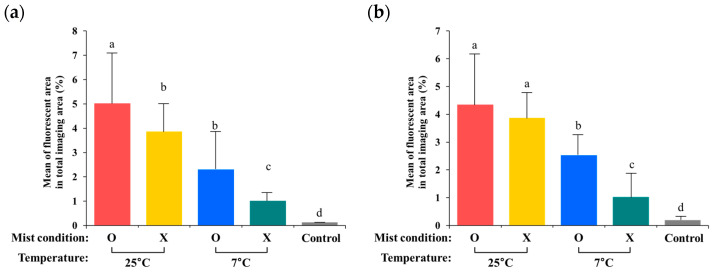
Mean fluorescent area (%) of internalized *S.* Typhimurium from day 1 to day 3 in iceberg lettuce (**a**), romaine lettuce (**b**), red lettuce (**c**), green onion (**d**), kale (**e**), and spinach (**f**). Significant differences (*p* < 0.05) are indicated with the lowercase letters of the alphabet.

**Table 1 foods-12-03106-t001:** Concentrations (mean, max, min, and SE) of internalized *S*. Typhimurium in iceberg lettuce, romaine lettuce, red lettuce, green onion, kale, and spinach (log CFU/g of fresh produce sample). The detection limit was 1.0 log CFU/g.

		Group 1 25 °C with Mist			Group 2 25 °C without Mist			Group 3 7 °C with Mist			Group 4 7 °C without Mist		
		Day1	Day2	Day3	Day1	Day2	Day3	Day1	Day2	Day3	Day1	Day2	Day3
Iceberg lettuce	Mean	3.2	2.9	2.9	2.8	2.8	2.6	2.5	2.1	2.0	1.7	1.7	1.7
	Max	4.2	3.5	3.5	3.3	2.8	3.3	3.2	2.7	2.4	2.1	2.3	2.2
	Min	2.3	2.4	2.5	2.3	2.1	2.2	2.1	1.6	1.3	1.0	1.3	1.1
	SE *	0.5	0.3	0.3	0.3	0.3	0.3	0.3	0.3	0.2	0.3	0.3	0.3
Romaine lettuce	Mean	3.0	2.7	2.7	2.6	2.5	2.4	2.2	2.0	1.9	1.9	1.9	1.8
	Max	3.3	3.1	3.2	3.1	3.0	3.0	2.6	2.5	2.1	2.4	2.8	2.2
	Min	2.5	2.1	2.2	1.9	2.0	1.8	1.7	1.4	1.6	1.3	1.3	0.9
	SE	0.2	0.3	0.3	0.3	0.3	0.3	0.2	0.3	0.1	0.3	0.4	0.3
Red lettuce	Mean	3.1	2.8	2.8	2.6	2.5	2.4	2.3	2.2	2.1	2.1	2.1	1.7
	Max	3.7	3.8	3.4	3.9	2.9	3.2	2.8	2.6	2.6	2.6	2.5	2.3
	Min	2.5	2.3	2.2	1.9	2.0	1.9	1.7	1.9	1.4	1.6	1.5	1.2
	SE	0.3	0.4	0.4	0.4	0.3	0.3	0.3	0.2	0.3	0.2	0.2	0.3
Green onion	Mean	1.9	1.8	1.8	1.5	1.2	1.2	1.4	1.2	1.2	1.5	1.3	1.2
	Max	2.5	2.2	2.2	2.0	1.8	1.9	1.9	1.9	2.1	1.8	1.9	1.9
	Min	1.1	0.9	0.9	0.8	0.6	0.6	1.0	0.5	0.9	0.6	0.9	0.5
	SE	0.4	0.3	0.4	0.3	0.4	0.4	0.3	0.4	0.4	0.4	0.3	0.4
Kale	Mean	2.2	2.2	1.7	1.7	1.6	1.6	1.3	1.6	1.5	1.5	1.5	1.5
	Max	2.5	2.5	2.1	2.0	2.1	2.0	1.8	2.0	2.1	2.0	2.0	2.0
	Min	1.9	1.8	1.2	1.3	1.0	1.1	0.8	1.2	1.0	0.9	1.0	0.9
	SE	0.2	0.2	0.3	0.2	0.3	0.2	0.3	0.2	0.3	0.3	0.3	0.3
Spinach	Mean	2.6	2.1	2.1	2.3	2.0	1.9	1.8	1.6	1.6	1.9	1.5	1.6
	Max	3.1	2.8	2.6	2.8	2.4	2.4	2.6	2.1	2.5	2.1	2.1	2.2
	Min	2.0	1.5	1.7	1.7	1.5	1.3	1.2	1.2	1.3	1.2	1.1	1.2
	SE	0.3	0.4	0.2	0.4	0.3	0.3	0.4	0.2	0.3	0.3	0.2	0.3

* SE: Standard error.

## Data Availability

Not applicable.

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
