# Peer review of "Internalization of Salmonella in Leafy Vegetables during Postharvest Conditions"

_foods, 2023, doi:10.3390/foods12163106_

Round 1

Reviewer 1 Report

This manuscript describes the effects of storage conditions (temperature and humidity) on internalization of bacteria Salmonella in six green leafy vegetables. The association of pathogens in food is one of the major causes of foodborne illness for human beings. Though many studies are available in this topic, a study of this kind will improve our existing knowledge on storage of foods. The authors addressed the research question by choosing appropriate research design. The paper is well written with supporting figures. The text is clear and readable. The results were analysed with appropriate statistical tests. The conclusions are supported by the observed results. The main question of the study is addressed with the observation of higher internalization of Salmonella at 25°C. Overall, the authors indicated that the Salmonella concentration was low at 7°C . While the research design and protocol used are acceptable, the authors need to provide more details on incubation of leafy vegetables. Specifically, the following points need to be addressed in the revision.
1. On what basis, the temperature range were used for this study?
2. How different temperature and humidity conditions were maintained throughout the incubation period
3. Provide the details of incubator

Author Response

REVISION NOTES

Reviewer 1

  1. On what basis, the temperature range were used for this study?

Response: In this study, the temperatures were 7℃ and 25℃. 7℃ is a common temperature for storing vegetables in grocery stores, while 25℃ was set as the temperature for room-temperature storage, such as traditional market.

  1. How different temperature and humidity conditions were maintained throughout the incubation period

Response: The treated samples were stored for 3 days in sterile plastic containers under four different conditions. We set up the temperature using a refrigerator for 7°C treated groups and room temperature for 25°C treated groups. The mist treatment was manually applied at predetermined intervals.

  1. Provide the details of incubator

Response: We updated the incubator information in the method section.

Revised text: For growing the GFP-labeled S. Typhimurium, Luria-Bertani (LB) medium with 100 µg/mL ampicillin (Sigma, St. Louis, MO, USA) was used in a shaking incubator (New Brunswick I2400 Incubator Shaker, Edison, NJ, USA) at 37°C.

Reviewer 2 Report

There are other similar works already exist and those are far better than this work. The article lacks any novelty. Abstract, introduction and other sections are reported very poorly. However, I suggest to you application of new approaches and methods for your works. Therefore, I suggest that the article be rejected.

This manuscript is still too poor with English and results are unclear. 

Author Response

Our study has similarities with previous research on bacterial internalization, but it also has new aspects that are detailed as follows:

1) Previous studies investigated bacterial internalization during plant cultivation, whereas we investigated bacterial internalization during storage after plant harvest for 3 days.

2) Previous studies examined the effects of temperature or humidity on bacterial internalization, while we investigated the combination effects of temperature (7°C vs 25°C) and humidity (non-mist vs mist conditions) on bacterial internalization.

3) Most studies investigated bacterial internalization in lettuce or a single crop, but we investigated bacterial internalization in various crops considering the characteristics of the plants.

4) Most previous studies used plate count methods to quantify bacterial internalization, while we newly introduced a method using a microscope for quantification of internalized bacteria.

We believe that our study presents new information on bacterial internalization and may provide fundamental information to protect public health and increase food safety.

We also revised our manuscript, including the abstract part, as reviewer’ comments. Would you please check the attached file?
Thank you so much for your comments.

Reviewer 3 Report

In this manuscript the authors are exploring the important subject of internalization of bacteria in fresh produce. While this has been shown to occur in the laboratory under various conditions, internalization has been questioned as a mechanism for field contamination. It is considered to be unlikely to occur in the field.

The laboratory studies here are interesting, but the way the bacteria is cultured and enumerated for all data points should be clarified (Table vs Figures).

The abstract states that internalization is a major contamination route and poses a public health risk. This statement is not true based on epidemiological data and scientific evidence. There is not enough data to support this and the word major should be replaced with possible and poses a potential public health risk.

This is stated in lines 37-38 and again this is not a valid statement. The references here (7 and 8) are lab studies and do not state that there is epidemiological evidence showing that internalization occurs or is a high risk for illness.

The authors should note that the references they discuss use relatively large numbers of bacteria under artificial circumstances.

Is there a reference for the cell density meter used in lines 76-77?

Line 82 should read 10e6 not 106.

How was the mist applied in the treatments and what is the final relative humidity?

The way the leaf samples were cut in lines 108-109 should be reworded and clarified.

Are the data in Table 1 from bacterial enumeration?

How do the authors distinguish if the results in Table 1 are bacteria attached to the edge or truly internalized. It has previously been shown that bacteria will preferentially bind to the cut edge of the lettuce leaf.  In Table 1 Romaine is misspelled.

In Table 1, a min and a max are given. Does this mean that all 10 samples were positive? Or were samples only prepared in triplicate?

There are a few places (noted in comments to author) where language should be clarified. 

Author Response

  1. The abstract states that internalization is a major contamination route and poses a public health risk. This statement is not true based on epidemiological data and scientific evidence. There is not enough data to support this and the word major should be replaced with possible and poses a potential public health risk.

This is stated in lines 37-38 and again this is not a valid statement. The references here (7 and 8) are lab studies and do not state that there is epidemiological evidence showing that internalization occurs or is a high risk for illness.

Response: Thank you so much for your comments and suggestion. I updated our manuscript in the abstract, introduction, and discussion (limitation of study) sections.

Revised text:

Abstract

Previous studies provided evidence that internalization of foodborne pathogens in fresh produce may be a potential contamination route and may poses a public health risk.

Introduction

Particularly, under laboratory conditions, internalization of foodborne pathogens in fresh produce may be one of the potential routes for contamination of fresh produce [7,8].

Discussion

Finally, this study on bacterial internalization was conducted in laboratory settings, so an epidemiological study is necessary.

  1. Is there a reference for the cell density meter used in lines 76-77?

Response: Yes, we added the reference.

Ge, C., Lee, C., & Lee, J. (2012). The impact of extreme weather events on Salmonella internalization in lettuce and green onion. Food Research International, 45(2), 1118-1122. 

  1. Line 82 should read 10e6 not 106.

Response: The error has been fixed.

Revised text: After making 1 × 106 CFU/mL of bacterial suspension with deionized water, 1 mL of the suspension was contaminated on surface of the lettuce samples (1 × 106 CFU/leaf). 

  1. How was the mist applied in the treatments and what is the final relative humidity?

Response: The information of the treatments has been updated in the method section.

Revised text: The mist treatment was applied for 1 minute (0.5 ml per minute) at predetermined intervals (every 2 hours) and final humidity in the container was approximately 75%. 

  1. The way the leaf samples were cut in lines 108-109 should be reworded and clarified.

Response: The text has been revised.

Revised text: For taking an image of internalized GFP-labeled Salmonella in the sample, the leafy parts of each sample were cut a piece (area ~1 to 2 cm2) of the middle periphery part of each sample using a sterile sampling knife (Fisher Scientific, WI, USA). 

  1. Are the data in Table 1 from bacterial enumeration?

Response: Yes, the data in the Table 1 from bacterial enumeration. We described the method in the section 2.1.

To remove surface bacteria (non-internalized bacteria) on the samples, our previous method was applied [6]. Briefly, the samples were emerged into 80% ethanol (Merck, Darmstadt, Germany) for 10 sec and then 1% AgNO3 (silver nitrate, Merck, Darmstadt, Germany) for 5 min, and finally rinsed with deionized water (Merck, Darmstadt, Germa-ny) for 10 sec. About 5 g of lettuce samples were homogenized with 20 mL of 0.1% peptone water (Thermo Fisher Scientific, Carlsbad, CA, USA) in a sterilized Whirl-Pak bag (Nasco, Fort Atkinson, WI, USA) for 5 min using a stomacher (Stomacher 80, Seward, West Sussex, UK). One hundred microliter of the suspension were spread on LB agar media supplemented with ampicillin (100 µg/mL) in triplicate. The LB agar plates were incubated at 37°C for 24 hours aerobically. Concentrations (colony) of the GFP-labeled S. Typhimurium were determined under UV light. In addition, ampicillin resistant gene was used for distinguishing targeted S. Typhimurium from other background bacterial flora. Ten samples of each group were examined and all experiments were conducted in triplicate. 

  1. How do the authors distinguish if the results in Table 1 are bacteria attached to the edge or truly internalized. It has previously been shown that bacteria will preferentially bind to the cut edge of the lettuce leaf. In Table 1 Romaine is misspelled.

Response: We removed surface bacteria (non-internalized bacteria) on the samples with 80% ethanol and AgNO3, which is a common method for study of bacterial internalization (Ge et al., 2012). In addition, we investigated bacterial contamination of leafy surface of the samples. Bacteria on the plant surface were not detected (<0 CFU/g of each plant). We described this information in our manuscript.

Revised text: After washing steps with ddH2O, AgNO3, and 80% ethanol solutions, bacteria on the plant surface were not detected (<0 CFU/g of each plant). 

  1. In Table 1, a min and a max are given. Does this mean that all 10 samples were positive? Or were samples only prepared in triplicate?

Response: Yest, all 10 samples were positive. There were 10 leafy samples of each group and each sample was subjected to three replicates of plate count method. The numbers of groups were 72 (=6 different type of fresh produce × 4 different treatment × 3 days). In the Table 1, values show the mean, minimum, maximum, and SE values of a total of 30 results. 

  1. There are a few places (noted in comments to author) where language should be clarified.

Response: we revised our manuscript.

Reviewer 4 Report

Comments are made directly on the manuscript.

Author Response

We revised the manuscript as your suggestion. Would you please check our updated manuscript and revision note?

Thank you so much for your kind comments.

REVISION NOTES

Reviewer 3

  1. Line 33: Please check the correctness of this reference for the statement

Response: Yes, we check these references. Thank you so much for your comments.

  1. Line 74: pellet

Response: The error has been fixed.

Revised text: The pellet of the GFP-labeled S. Typhimurium was collected using centrifugation at 6,500 g for 10 min and then resuspended using deionized water (Fisher).

  1. Lines 82: bracket close after mL

Response: The text has been revised as suggested.

Revised text: GFP-labeled S. Typhimurium was cultured in LB broth supplemented with ampi-cillin (10 µg/mL) for 18 hours at 37°C in a shaking incubator.

 Line 82, 84: as superscript

Response: The error has been fixed.

Revised text: After making 1 × 106 colony-forming unit (CFU)/mL of bacterial suspension with deionized water, 1 mL of the suspension was poured on surface of the lettuce samples (1 × 106 CFU/leaf).

 Line 83: suspension

Response: The text has been revised as suggested.

Revised text: After making 1 × 106 colony-forming unit (CFU)/mL of bacterial suspension with deionized water, 1 mL of the suspension was poured on surface of the lettuce samples (1 × 106 CFU/leaf).

 Line 84: poured/laid

Response: The text has been revised as suggested.

Revised text: After making 1 × 106 colony-forming unit (CFU)/mL of bacterial suspension with deionized water, 1 mL of the suspension was poured on surface of the lettuce samples (1 × 106 CFU/leaf).

 Line 101: harmonize throughout the manuscript

Response: The manuscript has been unified into "hours".

Revised text: The LB agar plates were incubated at 37°C for 24 hours aerobically.

 Line 138: Unit? Log CFU?

Response: We fixed the errors.

Revised text: Table 1. Concentrations of internalized S. Typhimurium in iceberg lettuce, romaine lettuce, red lettuce, green onion, kale, and spinach. (log CFU/g of sample)

 Table 1: very high concentration

Response: The error has been fixed. In addition, we checked the data once again.

  1. Table 1: In the method it is mentioned SEM not STD

Response: STD is stand for standard deviation. We marked the abbreviation.

Revised text: *STD: Standard deviation

  1. Line 148: after determined add- and presented

Response: The text has been revised as suggested.

Revised text: Concentrations of internalized S. Typhimurium in various fresh produce, including iceberg lettuce, romaine lettuce, red lettuce, green onion, kale, spinach, were deter-mined and presented in the Figure 1 and Table 1.

 Line 153: Is it the result of 7 C?

Response: It is a right sentence of the result of the 25°C with mist treated group. We edited the sentence.

Revised text: Particularly, under 25°C with mist condition, the mean concentration of internalized Salmonella in the iceberg lettuce sample was the highest and that in the green onion sample was the lowest (iceberg lettuce > red lettuce > romaine lettuce > spinach > kale > green on-ion).

  1. Line 159: Please check the comments in Figure 2

Response: The errors have been fixed.

  1. Line 167: SEM? Please correct throughout the text.

Response: We changed the text. In the manuscript, the abbreviation ‘SEM’ was used for ‘Scanning Electron Microscope’, so to avoid confusion, the ‘standard error of the mean’ was denoted as ‘SE’.

  1. Line 168: Please check the data again. Check the data in Table 1 for this treatment

Response: The error has been fixed.

Revised text: Concentrations (log CFU/g of iceberg lettuce) of internalized Salmonella Typhimurium in the iceberg lettuce samples (mean ± SE) were 3.0 ± 0.4 in the 25°C with mist condition; 3.5 ± 4.5 in the 25°C without mist condition; 2.2 ± 0.4 in the 7°C with mist condition; and 1.7 ± 0.3 in the 7°C without mist condition.

  1. Line 171: use- in after the word difference

Response: The grammatic errors have been fixed in the manuscript.

Revised text: However, no significant difference of Salmonella internalization was found between mist and non-mist groups (ANOVA, F = 0, p > 0.05).

  1. Line 176, 183, 190, 197, 204: SEM?

Response: We changed the text. In the manuscript, the abbreviation ‘SEM’ was used for ‘Scanning Electron Microscope’, so to avoid confusion, the ‘standard error of the mean’ was denoted as ‘SE’.

  1. Line 211-212: Temperature 27/25?

In the methodology it is mentioned 25 C (Page 4 Section 2.2).

In the distribution Triangular structure is evident. But in the legend only circular (Day 1 (Red)and day 2 (Yellow) and rectanglar structure is evident.  Please clarify.

Is it considering Day 1- Day 3: Fresh?

Response: Thank you so much for your comments. In this study, the temperature was set to 25°C, not 27°C. All typos have been corrected. While we did not confirm the freshness of the plants for three days in this study, we did not observe any significant changes visually. We will address this issue through additional research in the future. Furthermore, we described this information in the limitations of study in the discussion section.

  1. Line 222-223: or fresh produce was contaminated with Salmonella

Response: The text has been revised as suggested.

Revised text: However, in this study, fresh produce was contaminated with Salmonella at one time without additional contamination, and then trends of internalized Salmonella until 3 days were investigated.

  1. Line 227: Microscopic

Response: The error has been fixed.

Revised text: Microscopic images showing the topography of the surface of romaine lettuce and GFP-labeled S. Typhimurium.

  1. Line 236: 25 C?

Response: The error has been fixed.

Revised text: The groups stored at 25°C showed a larger fluorescent area (%) than the groups stored at 7°C.

 Line 240: Are these results for Day1/Day/2/Day3? Or Average of three days?

Response: We revised the legend of the Figure 3. These results showed average of three days (from day 1 to day 3).

Revised text: Figure 3. Mean fluorescent area (%) of internalized S. Typhimurium from day 1 to day 3 in iceberg lettuce (a), romaine lettuce (b), red lettuce (c), green onion (d), kale (e), and spinach (f). Significant differences (p<0.05) are indicated with the lowercase letters of the alphabet.

  1. Line 242: lowercase

Response: The text has been revised as suggested.

Revised text: Significant differences (p<0.05) are indicated with the lowercase letters of the alphabet.

  1. Line 259: Use reference serial number for these two references here

Response: The text has been revised.

Revised text: Ge et al., (2012) and Gu et al., (2013) demonstrated that internalization of Salmonella spe-cies in lettuce (2-3 log colony forming unit (CFU)/g) and tomato (0-3 log CFU/g)) occurred in seedling and mature plants [6,22].

  1. Line 261: Use reference serial number for this references here

Response: The text has been revised.

Revised text: Park et al., (2013) examined internalization of E. coli in spinach during a preharvest period [21].

  1. Line 269: add ) after e

Response: The error has been fixed.

Round 2

Reviewer 2 Report

The manuscript was improved. But, the manuscript contain little or no new information. Introduction and other sections are reported very poorly. No new application potentials of the work. Therefore, I suggest that the article be rejected.

Author Response

REVISION NOTES

  1. The manuscript was improved. But, the manuscript contain little or no new information. Introduction and other sections are reported very poorly. No new application potentials of the work. Therefore, I suggest that the article be rejected.

Response: Thank you for taking the time to review our manuscript and providing your valuable feedback. We appreciate your thoughts on the improvements made, as well as your concerns regarding the lack of new information, the poorly reported introduction and other sections, and the absence of new application potential. We understand that based on these issues, you are recommending rejection for our article. We would like to assure you that we take your comments seriously and will thoroughly reevaluate the manuscript, addressing the raised problems accordingly. We modified our entire manuscript, including introduction, method and materials, results, and conclusion.